# Study on Characteristics of Failure and Energy Evolution of Different Moisture-Containing Soft Rocks under Cyclic Disturbance Loading

**DOI:** 10.3390/ma17081770

**Published:** 2024-04-12

**Authors:** Xuewen Cao, Xuhui Tang, Lugen Chen, Dong Wang, Yujing Jiang

**Affiliations:** 1College of Energy and Mining Engineering, Shandong University of Science and Technology, Qingdao 266590, China; xwcao0903@163.com (X.C.); lgchen854@163.com (L.C.); wdwinter@163.com (D.W.); jiang@nagasaki-u.ac.jp (Y.J.); 2Infrastructure Department, Shandong University of Science and Technology, Qingdao 266590, China; 3State Key Laboratory of Mining Disaster Prevention and Control Co-Founded by Shandong Province and the Ministry of Science and Technology, Shandong University of Science and Technology, Qingdao 266590, China

**Keywords:** cyclic disturbance, failure characteristics, energy evolution, soft rock, moisture content

## Abstract

During the coal mining process in soft rock mines with abundant water, the rock mass undergoes cyclic loading and unloading at low frequencies due to factors such as excavation. To investigate the mechanical characteristics and energy evolution laws of different water-containing rock masses under cyclic disturbance loading, a creep dynamic disturbance impact loading system was employed to conduct cyclic disturbance experiments on various water-containing soft rocks (0.00%, 1.74%, 3.48%, 5.21%, 6.95%, and 8.69%). A comparative analysis was conducted on the patterns of input energy density, elastic energy density, dissipated energy density, and damage variables of different water-containing soft rocks during the disturbance process. The results indicate that under the influence of disturbance loading, the peak strength of specimens, except for fully saturated samples, is generally increased to varying degrees. Weakness effects on the elastic modulus were observed in samples with 6.95% water content and saturated samples, while strengthening effects were observed in others. The input energy density of samples is mostly stored in the form of elastic strain energy within the samples, and different water-containing samples adapt to external loads within the first 100 cycles, with almost identical trends in energy indicators. Damage variables during the disturbance process were calculated using the maximum strain method, revealing the evolution of damage in the samples. From an energy evolution perspective, these experimental results elucidate the fatigue damage characteristics of water-containing rock masses under the influence of disturbance loading.

## 1. Introduction

With the prolonged and continuous exploitation of coal resources in China, accessible coal reserves are steadily diminishing, and coal mining activities are progressively shifting towards deeper deposits [1,2,3]. As coal resources are extracted, the dynamic stress adjustment in the roof and floor of goaf zones can be regarded as subject to perturbation loading. Additionally, deep mines are often subjected to mechanical vibrations and fatigue loads from activities such as blasting [4,5,6,7]. In coal mines with rich water-bearing soft rocks, the presence of water can deteriorate the rock properties [8,9,10], thereby weakening the stability of mine roadway surroundings. Therefore, investigating the damage and energy evolution characteristics of rocks under the combined action of dynamic–static cyclic loading and water presents significant engineering implications and theoretical value.

Scholars have carried out compression [11,12,13], tension [14], and fracture [15,16] experiments that reveal the mechanical properties and fracture characteristics of various types of rocks under dynamic loading conditions. Jiang et al. [17] studied the impact of small-amplitude cyclic dynamic perturbation on soft rock–coal dual materials, detected the acoustic emission characteristic parameters of the sample during the loading process, and analyzed its main frequency characteristics. Dehghanipoodeh et al. [18] studied the mechanical properties and deformation characteristics of different grouting rocks under static and dynamic loads. Malik et al. [19] conducted dynamic compression experiments on basalt using a separated Hopkinson rod and proposed an empirical correlation of the dynamic increase factor of compressive strength. Dong et al. [20] examined the mechanical properties of sandstone and found a nonlinear relationship between sample compressive strength and initial cyclic values. Arora et al. [21] used sine wave loading to conduct cyclic compression tests on different types of rocks and concluded that the rock secant modulus degradation is related to the number of cycles and proposed a normalized relationship for rock modulus degradation.

A substantial body of experimental research has been carried out on the weakening effects of water on rocks and yielded a wealth of findings [22,23]. Perera et al. [24] investigated the strength and deformation characteristics of coal rocks in their natural and saturated states. Feng et al. [25] studied the damage behavior of sandstone under the joint influence of moisture content and intermediate principal stress and concluded that with the increase in moisture content and intermediate principal stress, tensile cracks tend to increase, while shear cracks gradually reduce. Sun et al. [26] carried out creep experiments with different moisture conditions for sandstone in the roof strata of the Wanfu coal mine, providing a reliable theoretical basis for early warning of roadway creep failure. Zhang et al. [27] conducted uniaxial and cyclic loading and unloading experiments on rocks in dry, unsaturated, and saturated states, and obtained the changing rules of mechanical characteristic parameters of sandstone in different water-containing states.

Simultaneously, rock deformation is accompanied by the absorption and release of energy, constituting a damage process driven by energy considerations [28]. Meng et al. [29] conducted cyclic loading and unloading experiments on rocks with different loading rates and obtained the energy evolution characteristics of rocks with different lithologies. Tarasov et al. [30] discussed the role of elastic energy in determining the dynamic energy balance and the fracture mechanism operating during spontaneous failure during loading of different brittle rocks, which provides a certain theoretical basis for understanding the dynamic process of rock bursts. Hu et al. [31] revealed the energy evolution law of weakly cemented sandstone under the action of creep, which provided a certain theoretical basis for the long-term stability of weakly cemented surrounding rock mine tunnels.

In summary, the current analysis of the failure modes and physical and mechanical properties of water-bearing rocks under the action of cyclic disturbance loads mainly focuses on dry, natural, and saturated states. There is a lack of quantitative analysis of the damage and destruction of water-bearing soft rocks under the action of perturbation loads. From a mechanical point of view, the rock deformation and failure process is a process from local destruction to overall catastrophe, and this process is accompanied by energy input, accumulation, and dissipation. Thermodynamics believes that rock failure is an instability phenomenon driven by energy. Studying materials from an energy perspective can better reveal the essential characteristics of their failure. Therefore, studying loaded rocks from an energy perspective and analyzing, in detail, the energy evolution rules during the deformation and failure process of water-rich soft rocks can provide new ideas for the analysis and prediction of deep soft rock engineering disasters caused by rock instability.

## 2. Materials and Experimental Methods

### 2.1. Materials

The specimens were sourced from the silty sandstone in the Shanghaimiao mining area of Ordos, Inner Mongolia. Following the standards set by the International Society for Rock Mechanics (ISRM), the samples underwent three processes of coring, cutting, and grinding to produce standard specimens with a diameter of 50 mm and a height of 100 mm. The samples exhibited a yellowish color and a dense structure. A visual inspection was carried out and specimens with no visible bedding, streaks, or cracks, and with excellent overall integrity and uniformity, were selected as the standard specimens. Figure 1 shows a few of the experimental samples.

The preparation steps for sandstone samples with varying moisture content are as follows: (1) All prepared samples were initially placed in a drying oven and dried at a temperature of 106 °C for 24 h. After drying, the samples were weighed. When the mass remained essentially unchanged in two consecutive weighings, the drying process was considered complete, and the dried mass was recorded. (2) Samples, excluding those designated for drying, were immersed in distilled water. Then, the samples were removed at intervals, their surface moisture was wiped off, and their masses were recorded. The saturation point was considered to have been reached when the mass remained constant within adjacent 12 h periods. (3) Based on the magnitude of saturation, the moisture content for the intermediate four gradients was calculated, and at each level, the corresponding sample masses were inferred. The samples were immersed in distilled water, and measurement intervals were reduced as their masses approached the desired moisture content level. At such point, the preparation of unsaturated samples was completed.

### 2.2. Experimental Equipment and Protocol

The experimental equipment for this experiment is the creep impact dynamic disturbance loading system independently developed by Shandong University of Science and Technology, as shown in Figure 2. This system is capable of applying both axial static and dynamic loads. The static load unit has a maximum capacity of 800 kN, while the dynamic load unit can handle a maximum of 100 kN. The system can apply complex waveforms, including sinusoidal, rectangular, and custom waveforms. The latter part requires stress, displacement, and loading path waveforms to be independently designed. The perturbation waveform frequency ranges from 0.01 to 10.00 Hz. Data are sampled at intervals of 0.05 s during static loading, while sampled at intervals of 0.001 s for dynamic loading. Displacement is measured using Demec magnetic incremental displacement sensors with a range of up to 200 mm and an accuracy of 0.002 mm. Throughout the experiment, the system continuously collects data on axial load, axial strain, and time, and records corresponding parameter curves in a specified directory. The experiment is divided into two parts. The first part involves determining the static mechanical parameters of different moisture content samples and the initial value of σ_m_ for cyclic loading by using uniaxial compression tests. The second part entails cyclic loading experiments under certain static loading conditions, and the experimental plan is detailed in Table 1. The samples are first loaded to a specific static load σ_m_ at a constant displacement rate of 0.1 mm/min using a closed-loop, constant velocity displacement control testing machine. Subsequently, with σ_m_ as the average static load, periodic cyclic dynamic loads are applied to the samples. To simulate the elastic waves during water drainage and energy storage in mining and seismic vibrations, we selected a sinusoidal waveform for the cyclic perturbation waveform with a frequency of 5 Hz and an amplitude of 10 kN. The perturbation cycles were then set at 1000. Figure 2 provides a schematic representation of the sample loading path.

## 3. Analysis of Experimental Results

### 3.1. Mechanical Properties Analysis

Figure 3 shows the stress-strain curve of the sample under static loading conditions, the peak strengths of different moisture content samples decrease by level as moisture content increases, indicating different softening tendencies. After reaching the peak, dry samples display linear stress reduction and exhibit brittle fracture characteristics. By contrast, moist samples show a gradual slowing of stress reduction with curves displaying varying degrees of stress recovery, indicating ductile characteristics. With increasing moisture content, the peak strengths of the samples and the slope of the stress–strain curve gradually decrease. Figure 4 illustrates the trends in average peak strength and elastic modulus as a function of moisture content, demonstrating the significant influence of water on the mechanical properties of the rock samples. As moisture content increases, the average uniaxial compressive strengths of the samples at moisture levels of 0.00%, 1.74%, 3.48%, 5.21%, 6.95%, and 8.69% decrease to 32.66, 17.93, 15.93, 15.52, 13.91, and 12.98 MPa, respectively, representing a continuous decline in strength. The corresponding elastic moduli are 4.03, 2.92, 2.70, 2.64, 2.37, and 2.30 GPa, showing a similar reduction trend. The initial value of 32.66 MPa compressive strength decreases by 60.26%, resulting in a softening coefficient of 0.397, while the elastic modulus initial value of 4.03 GPa decreases by 42.93%, yielding a reduction coefficient of 0.571. Both uniaxial compressive strength and elastic modulus for samples at different moisture levels exhibit an exponential decrease in relation to moisture content. The relationship between these variables is mathematically expressed in Figure 4.

In addition to being subjected to axial stress in deep water-rich soft rocks, the mines are also exposed to periodic perturbation loading such as mechanical vibrations and mine-induced seismic events. In this study, a series of cyclic perturbation experiments, as outlined in Table 1, were carried out on multiple sets of samples with varying levels of moisture content. Figure 5 shows the resulting stress–strain curves for the samples. The trend of perturbation experiment curves follows a pattern similar to the uniaxial compression curves, with both showing brittle characteristics for dry samples and ductile characteristics for moist samples after the peak. However, after the perturbation tests, peak strength and elastic modulus show significant differences compared with uniaxial compression. Figure 6 illustrates the trends in average peak strength and elastic modulus as a function of moisture content, highlighting the substantial influence of moisture on the mechanical properties of rock samples. With increasing moisture content, the perturbation strength gradually decreases from an initial 35.87 to 12.56 MPa, which is a decrease of 64.98%. Similarly, the elastic modulus decreases from 4.55 to 1.92 GPa, a reduction of 57.80%. In comparison to uniaxial compression under dynamic perturbation loading, the peak strength of the samples increases by 9.83%, 15.78%, 11.99%, 6.37%, and 2.73% at moisture levels of 0.00%, 1.74%, 3.48%, 5.21%, and 6.95%, respectively, and decreases by 2.54% at a moisture level of 8.69%. The elastic modulus increases by 12.90%, 10.96%, 4.40%, and 0.75% at moisture levels of 0.00%, 1.74%, 3.48%, and 5.21%, respectively, and decreases by 14.35% and 16.52% at moisture levels of 6.95% and 8.69%. This behavior can be attributed to the internal particles becoming less rigid when losing water in a dry environment, increasing the friction between particles. Lower-amplitude perturbations result in recompacting internal voids, to a certain extent increasing the sample’s elastic modulus. As the moisture content of the sample increases, the water in the internal pores relatively increases. Under cyclic perturbation loading, several pores lose their load-bearing capacity, leading to instability and failure, which reduce the strengthening effect. In full saturation state, the pores within the sample are filled with water. On one hand, water provides a degree of lubrication, but on the other hand, it is an incompressible fluid. During the cyclic perturbation, the process causes changes in pore water pressure that, in turn, lead to pore expansion and interconnection. Thus, the compressive strength and elastic modulus decrease.

### 3.2. Macro-Destructive Features

Figure 7 shows the typical failure modes of different moisture content samples under cyclic loading. We observe that for dry samples, cracks initiate from the top and extend towards the middle, representing a typical tensile failure mode. As the moisture content increases, the failure mode transitions from pure tensile to a tensile-shear composite failure, with shear cracks becoming more pronounced with increasing moisture content. When the moisture content reaches 5.21%, a clear “X”-shaped shear failure trend is observed. Further increasing the moisture content to 6.95% increases the prominence of the “X”-shaped shear failure mode. When the sample reaches a saturated state, the internal voids are filled with water, weakening the friction between particles and providing lubrication. At this point, the sample exhibits a single inclined plane shear failure. Thus, under the influence of cyclic perturbation loading, the macroscopic failure mode of the sample evolves from tensile failure to a tensile-shear composite failure and eventually to a single inclined plane shear failure.

### 3.3. Characteristics of Energy Evolution

It is expected that studying the damage characteristics of the surrounding rock mass of water-rich soft rock mines under the action of cyclic dynamic loads from an energy perspective will aid in better describing the essential characteristics of deep soft rock instability. It is also important for understanding the damage energy mechanism and dynamic disasters of rock masses under cyclic dynamic loads. The development of topics such as mechanisms is of positive significance.

The total input energy (U) generated by the external forces performing at work is the sum of elastic energy ($U^{e}$) and dissipative energy ($U^{d}$) (assuming there is no heat exchange occurring with the surroundings during the physical process). In uniaxial cyclic loading–unloading experiments, the stress–strain curve of the sample and the area enclosed by the coordinate axes represent the energy variation characteristics of the sample during the loading process. As shown in Figure 8, the loading curve AB and the area enclosed by the coordinate axes represent the total energy density (U) absorbed by the rock sample, which is the work carried out by the testing machine on the rock sample during loading. The unloading curve BC and the area enclosed by the coordinate axes represent the accumulated elastic energy density ($U^{e}$) stored within the rock sample and can be released during the unloading process. The dissipative energy ($U^{d}$) of the rock sample can be calculated as the difference between these two areas, namely, those enclosed by ABC and the coordinate axes. This portion of energy is typically used to overcome internal damping effects, friction, plastic deformation, and other factors. In the following formulas, $\sigma_{i}^{+}$ represents the loading curve for the i-th cycle, $\sigma_{i}^{-}$ represents the unloading curve for the *i*-th cycle, and $\sigma_{i+1}^{+}$ represents the loading curve for the (*i* + 1)-th cycle. The formulae for calculating each energy indicator are as follows:(1)$$U^{i}=U^{ei}+U^{di}$$
(2)$$U^{i}=\int_{\varepsilon_{A}}^{\varepsilon_{B}}\sigma_{i}^{+}d\varepsilon_{i}$$
(3)$$U^{ei}=\int_{\varepsilon_{c}}^{\varepsilon_{B}}\sigma_{i}^{-}d\varepsilon_{i}$$
(4)$$U^{di}=U^{i}-U^{ei}=\int_{\varepsilon_{A}}^{\varepsilon_{C}}\left(\sigma_{i}^{+}-\sigma_{i}^{-}\right)d\varepsilon_{i}$$
where $U^{i}$, $U^{ei}$, and $U^{di}$ denote the input energy density, elastic energy density, and dissipated energy density in the *i*-th cycle, respectively; $\sigma_{i}^{+}$ and $\sigma_{i}^{-}$ denote, the stresses on the loading and unloading curves in the *i*-th cycle, respectively; and $\varepsilon_{A}$, $\varepsilon_{B}$, and $\varepsilon_{C}$ denote the strains, as shown in Figure 9.

The fatigue damage variable *D* is an important parameter used to describe the extent of damage to the sample under fatigue loading. Xiao et al. [32] suggested that the fatigue variable can be defined using the maximum strain method, as follows:(5)$$D=\frac{\varepsilon_{max}^{n}-\varepsilon_{max}^{0}}{\varepsilon_{max}^{f}-\varepsilon_{max}^{0}}$$
where $\varepsilon_{max}^{0}$ is the initial cyclic maximum strain value, $\varepsilon_{max}^{n}$ is the instantaneous maximum strain value after n cycles, and $\varepsilon_{max}^{f}$ is the limiting strain value during the disturbance.

Using Equations (1)–(4), the fatigue energy parameters of water-containing rock samples under dynamic disturbance loading are determined. The parameters include input energy density, elastic energy density, and dissipative energy density. Figure 9 shows the representative curves of energy density parameters for different water-containing soft rocks under cyclic disturbance loading, and the corresponding results are presented in Table 2, Table 3 and Table 4. From the graph, we observe that the input energy density of the water-containing samples decreases to varying degrees from the 1st to the 100th cycle, while that of saturated samples remains almost constant. With the same number of disturbance cycles, the input energy density of the samples gradually decreases as the water content increases. For example, in the first cycle, between adjacent water contents (0.00% and 1.74%, 1.74% and 3.48%, and so on), the input energy density of the samples decreases by 29.88%, 7.05%, 3.03%, 9.47%, and 5.98%, respectively. The reason is that the initial values of the disturbance cycles are selected in the linear elastic stage of the samples during the experiment. At this stage, the inherent cracks and micropores in the samples are compacted. However, during the initial phase of the disturbance cycles, the load with an amplitude of 10 kN sudden change occurs, causing the originally compacted microcracks and micropores inside the samples to reach their bearing limits and fail. As such, the samples show significant deformation during the first cycle, leading to increased input energy. As the water content further increases, the amount of water increases inside the samples, leading to an increase in softening and a decrease in elastic modulus. However, the presence of water has a certain buffering effect, thereby weakening the impact of load fluctuations on the samples, resulting in smaller deformations and lower input energy. As the disturbance cycles continue, the elastic energy density of the samples increases initially and then stabilizes, and gradually decreases with an increase in water content. For example, in the first cycle, between adjacent water contents (0.00% and 1.74%, 1.74% and 3.48%, and so on), the elastic energy densities of the samples decrease by 31.38%, 5.93%, 3.80%, 11.70%, and 5.37%, respectively. The reason is a load fluctuation during the disturbance stage, and part of the energy is used to develop new microcracks and micropores. After 100 disturbances, the internal stresses in the samples are readjusted, causing recompaction of the newly developed microcracks and micropores. The samples adapt to external disturbance loads, leading to a stable mechanical performance. Elastic energy density accounts for over 85% of input energy density and gradually decreases with increasing water content, from 89.76% in the dry state to 86.59% in the saturated state. This result indicates that during dynamic disturbance, energy is mainly stored as elastic energy within the rock, with only a small fraction used for overcoming material damping and plastic deformation. The evolution of dissipative energy density follows the trend of elastic energy density, initially decreasing and then stabilizing, gradually decreasing with increasing water content. This reason is the load fluctuation effect, which causes previously closed microcracks and micropores to reach their bearing limits, increasing the higher dissipative energy from the 1st to the 100th disturbance.

The damage variable D for different water-containing soft rocks under dynamic disturbance is calculated using Equation (5), as shown in the figure. A comparison of the graphs reveals that the change in disturbance-induced damage variable D for different water-containing soft rocks follows a nearly identical trend, characterized by two phases: an accelerating and a stable accumulation. Given that most of the damage to the samples occurs within the first 100 disturbance cycles, the damage variable D from the 1st to the 100th cycle exhibits an accelerating accumulation state and gradually adapts to the external disturbance load. After 100 disturbances, the internal crack structure redistributes and stabilizes, thereby entering a phase of stable damage.

## 4. Discussion

The surrounding rock mass of deep water-rich soft rock mine tunnels is disturbed by excavation, mining, etc., and the stress state changes. Energy change is the essential characteristic of physical changes in materials. This paper uses energy as the basis to explore the mechanical response behavior and internal damage rules of different water-bearing soft rocks under cyclic disturbance, and determines the saturated moisture content of the samples. Through further experiments, the mechanical parameters and macroscopic damage characteristics of different samples were obtained, the changes in mechanical parameters of the samples under the action of disturbance factors were analyzed, and the evolution rules of the energy of the samples with the number of disturbances and moisture content during the disturbance process were obtained through calculations. By analyzing the data obtained, it was found that the disturbance load has a certain strengthening effect on samples with low moisture content, and this strengthening effect gradually decreases as the moisture content increases. During the process of cyclic perturbation, all samples adapted to external loads within 100 cycles. In subsequent cycles, the energy changes of the samples tended to a constant value, and most of the energy was stored inside the rock samples in the form of elastic energy. This also leads to the shortcoming that the damage characteristics of the sample are not obvious during the disturbance stage. In this experiment, the initial value of the cyclic disturbance σ_m_ selected for the disturbance load did not exceed the upper limit stress value of the sample, and the sample was not damaged during the disturbance. As a result, the energy evolution law of the sample when it was damaged by the disturbance load was not detected during the experiment. The experiment itself also has limitations and shortcomings, which are specifically reflected by the following: (1) Only the initial value of the cyclic disturbance load is considered, while the disturbance amplitude and frequency are ignored; (2) There is a lack of long-term simulation of the surrounding rock under real underground disturbance loads. This experiment only explored the mechanical behavior under short-term disturbance. These shortcomings need to be carefully considered in future studies to obtain more comprehensive and accurate results.

## 5. Conclusions

(1)Mudstone is significantly influenced by its water content. The uniaxial compressive strength of mudstone decreases from 32.66 MPa to 12.98 MPa, resulting in a softening coefficient of 0.397. The elastic modulus decreases from 4.03 GPa to 2.30 GPa, with a coefficient of 0.571. In the samples, peak strength and elastic modulus exhibit an exponential decrease with increasing water content.(2)In cyclic disturbance experiments, compressive strengths of unsaturated samples generally increase to varying degrees, while saturated samples decrease in strength. The elastic modulus of the samples shows a strengthening effect under cyclic disturbance loading at 6.95% in the saturated water state but exhibits a weakening effect in other cases. Cyclic disturbance loading does not change the failure mode of the samples; that is, dry samples become brittle while wet samples fail in a plastic manner.(3)All samples adapt to external disturbance loading within the first 100 cycles and then maintain relative stability. An inverse relationship is observed between energy indicators and water content. The variation trends of energy indicators for different water-containing samples are nearly identical, with a turning point occurring near 100 cycles. During the cyclic disturbance loading, energy is mainly stored as elastic energy within the samples. The evolution of damage in different water-containing soft rocks can be divided into two, an accelerating and a stable accumulation phase. Damage accumulation accelerates from the 1st to the 100th cycle, then stabilizes after 100 cycles.

## Figures and Tables

**Figure 1 materials-17-01770-f001:**
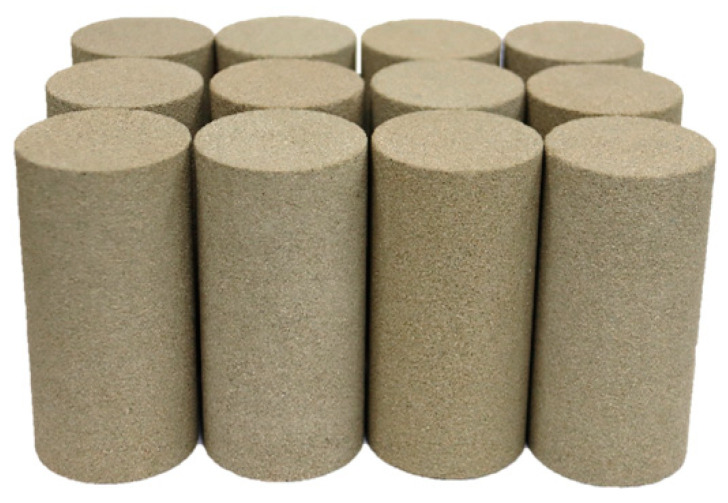
Muddy sandstone specimen.

**Figure 2 materials-17-01770-f002:**
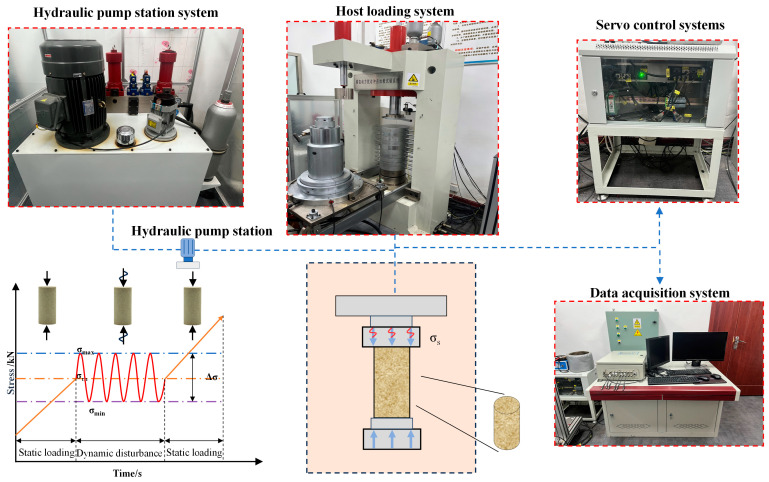
Schematic diagram of creep dynamic disturbance impact loading system.

**Figure 3 materials-17-01770-f003:**
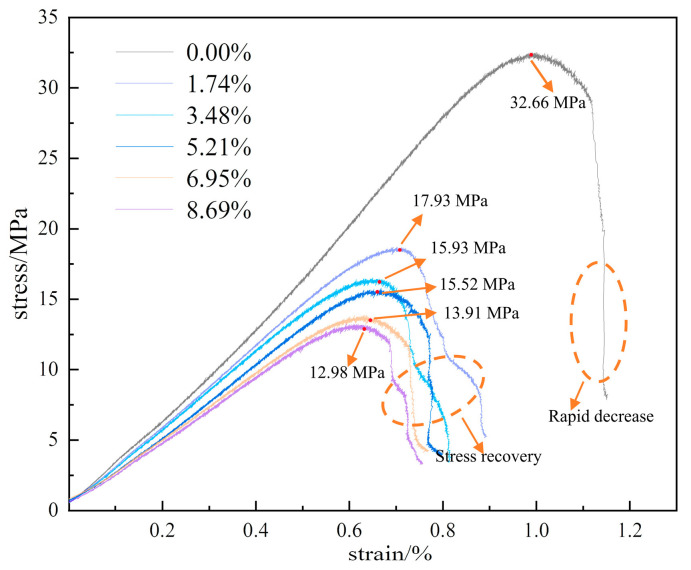
Uniaxial compressive stress–strain curves.

**Figure 4 materials-17-01770-f004:**
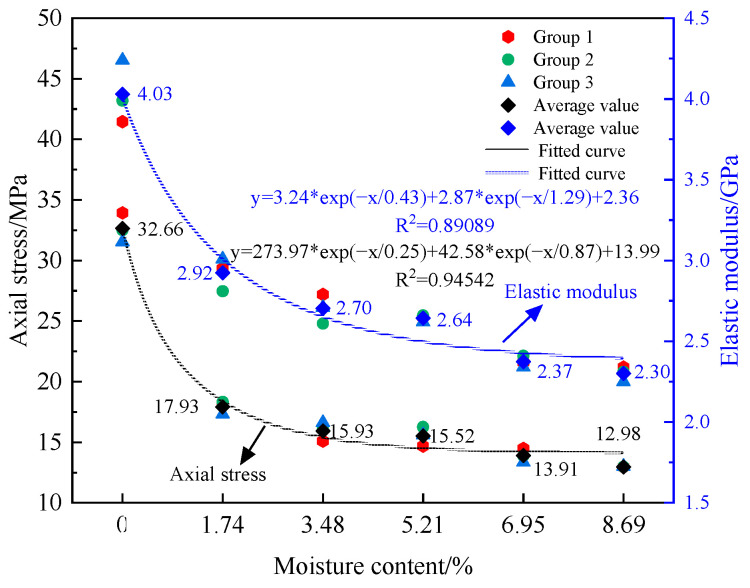
Mechanical parameter change curve.

**Figure 5 materials-17-01770-f005:**
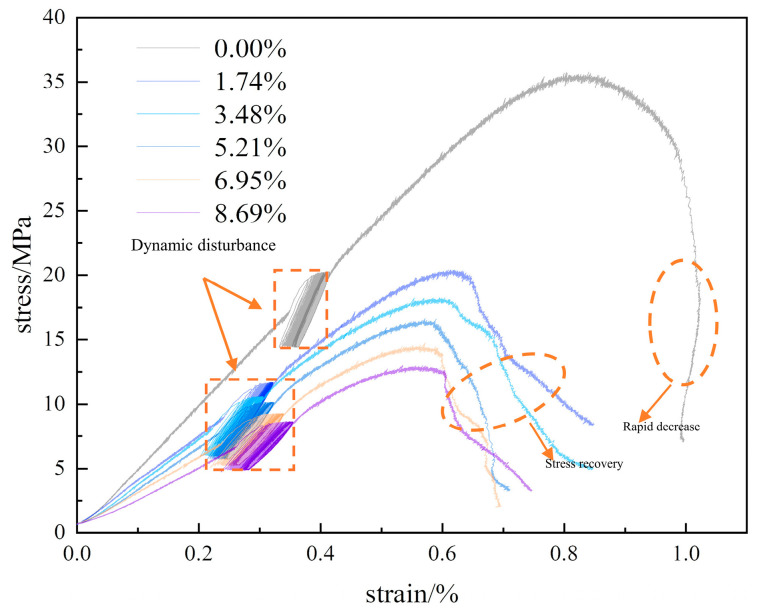
Cyclic disturbance stress–strain curve.

**Figure 6 materials-17-01770-f006:**
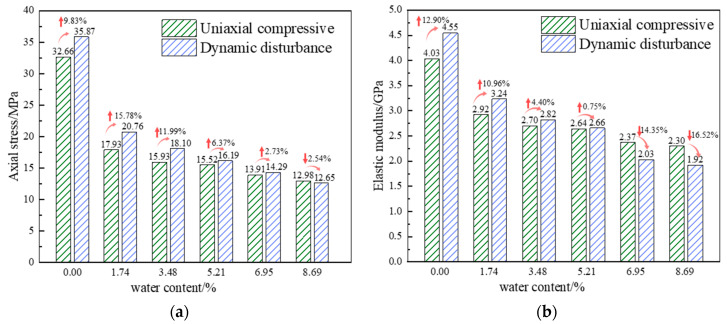
Changing pattern of mechanical properties of the samples: (**a**) Peak strength, (**b**) modulus of elasticity.

**Figure 7 materials-17-01770-f007:**
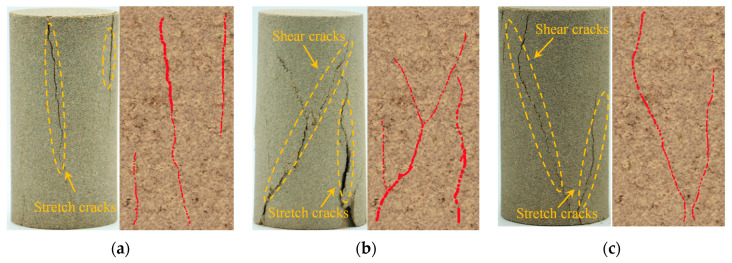
Macroscopic damage characteristics of samples with different water contents. (**a**) 0.00%; (**b**) 1.74%; (**c**)3.48%; (**d**) 5.12%; (**e**) 6.95%; (**f**) 8.69%.

**Figure 8 materials-17-01770-f008:**
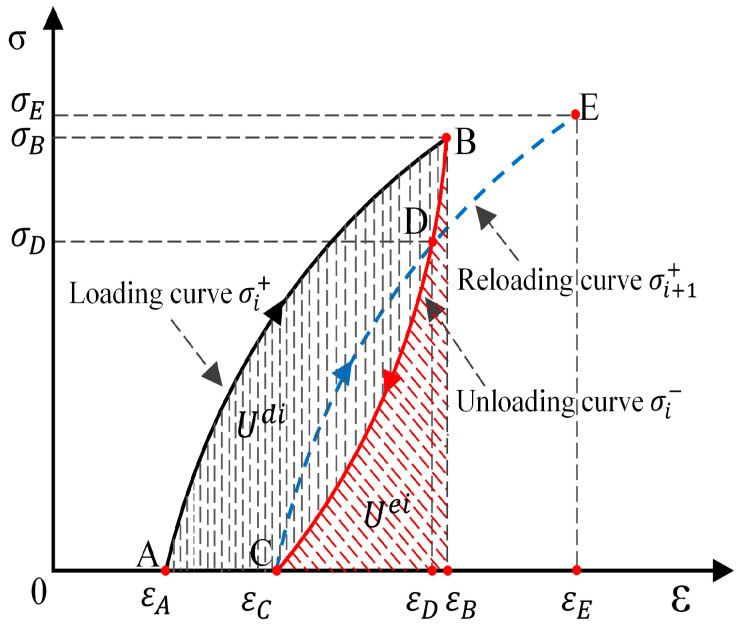
Schematic diagram of cyclic loading energy calculation.

**Figure 9 materials-17-01770-f009:**
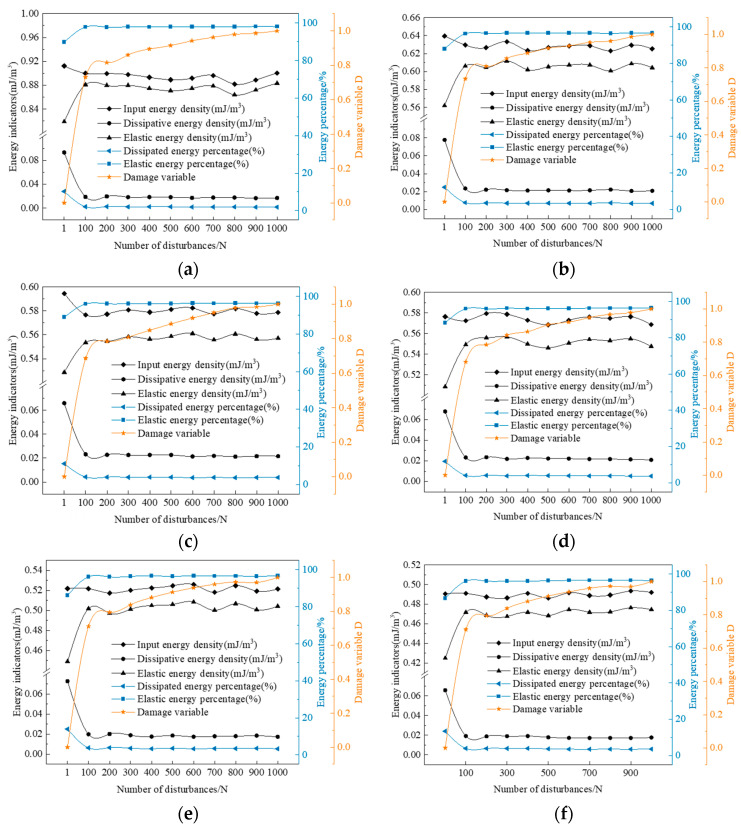
Characteristics of energy evolution of different water-bearing samples: (**a**) 0.00%, (**b**) 1.74%, (**c**) 3.48%, (**d**) 5.21%, (**e**) 6.95%, (**f**) 8.69%.

**Table 1 materials-17-01770-t001:** Loading scheme for perturbation experiments.

Rock Sample Number	Average Water Content (%)	σ_m_ (MPa)	Δσ (kN)	Frequency (Hz)	Number of Disturbances/N
R0-1	0	16.33	10	5	1000
R0-2			10	5	1000
R0-3			10	5	1000
R2-1	1.74	8.97	10	5	1000
R2-2			10	5	1000
R2-3			10	5	1000
R4-1	3.48	7.97	10	5	1000
R4-2			10	5	1000
R4-3			10	5	1000
R6-1	5.21	7.76	10	5	1000
R6-2			10	5	1000
R6-3			10	5	1000
R8-1	6.95	6.96	10	5	1000
R8-2			10	5	1000
R8-3			10	5	1000
R10-1	8.68	6.49	10	5	1000
R10-2			10	5	1000
R10-3			10	5	1000

**Table 2 materials-17-01770-t002:** Input energy density (mJ/m^3^) of water-bearing samples with different number of perturbations.

Moisture Content (%)	Number of Disturbances (N)										
	1	100	200	300	400	500	600	700	800	900	1000
0.00	0.9122	0.8998	0.8994	0.8980	0.8932	0.8891	0.8918	0.8963	0.8817	0.8886	0.9004
1.74	0.6396	0.6296	0.6266	0.6333	0.6234	0.6269	0.6286	0.6288	0.6230	0.6294	0.6252
3.48	0.5945	0.5767	0.5773	0.5808	0.5789	0.5813	0.5824	0.5776	0.5819	0.5778	0.5787
5.21	0.5765	0.5726	0.5795	0.5788	0.5729	0.5685	0.5729	0.5761	0.5749	0.5764	0.5687
6.95	0.5219	0.5218	0.5174	0.5201	0.5224	0.5247	0.5258	0.5181	0.5248	0.5193	0.5215
8.69	0.4907	0.4909	0.4874	0.4864	0.4909	0.4862	0.4917	0.4888	0.4893	0.4934	0.4920

**Table 3 materials-17-01770-t003:** Dissipative energy density (mJ/m^3^) of water-bearing samples with different number of perturbations.

Moisture Content (%)	Number of Disturbances (N)										
	1	100	200	300	400	500	600	700	800	900	1000
0.00	0.0934	0.0189	0.0197	0.0184	0.0186	0.0183	0.0172	0.0177	0.0178	0.0167	0.0171
1.74	0.0777	0.0236	0.0222	0.0216	0.0213	0.0215	0.0213	0.0215	0.0223	0.0208	0.0209
3.48	0.0659	0.0232	0.0227	0.0225	0.0225	0.0226	0.0213	0.0218	0.0212	0.0216	0.0216
5.21	0.0680	0.0233	0.0235	0.0219	0.0229	0.0223	0.0221	0.0218	0.0217	0.0214	0.0210
6.95	0.0729	0.0199	0.0202	0.0189	0.0175	0.0187	0.0174	0.0179	0.018	0.0185	0.0173
8.69	0.0658	0.0191	0.0188	0.0189	0.0191	0.0178	0.0172	0.017	0.0171	0.0171	0.0175

**Table 4 materials-17-01770-t004:** Elastic energy density (mJ/m^3^) of water-bearing samples with different number of perturbations.

Moisture Content (%)	Number of Disturbances (N)										
	1	100	200	300	400	500	600	700	800	900	1000
0.00	0.8188	0.8809	0.8797	0.8796	0.8746	0.8708	0.8746	0.8786	0.8639	0.8719	0.8833
1.74	0.5619	0.606	0.6044	0.6117	0.6021	0.6054	0.6073	0.6073	0.6007	0.6086	0.6043
3.48	0.5286	0.5535	0.5546	0.5583	0.5564	0.5587	0.5611	0.5558	0.5607	0.5562	0.5571
5.21	0.5085	0.5493	0.5560	0.5569	0.5500	0.5462	0.5508	0.5543	0.5532	0.5550	0.5477
6.95	0.4490	0.5019	0.4972	0.5012	0.5049	0.5060	0.5084	0.5002	0.5068	0.5008	0.5042
8.69	0.4249	0.4718	0.4686	0.4675	0.4718	0.4684	0.4745	0.4718	0.4722	0.4763	0.4745

## Data Availability

Data associated with this research are available and can be obtained by contacting the corresponding author upon reasonable request.

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
