# Peer review of "Study on Characteristics of Failure and Energy Evolution of Different Moisture-Containing Soft Rocks under Cyclic Disturbance Loading"

_materials, 2024, doi:10.3390/ma17081770_

Round 1

Reviewer 1 Report

Comments and Suggestions for Authors

Journal:  Materials (ISSN 1996-1944)

Manuscript ID: materials-2918547

Study on Characteristics of Failure and Energy Evolution of Different Moisture-Contenting Soft Rocks under Cyclic Disturbance Loading, by Xuewen Cao, Xuhui Tang, Lugen Chen, Dong Wang, Yujing Jiang

The authors described the experiments well and processed the experimental results well. In the conclusion, they summarized all the results obtained by analyzing the experimental results. However, the abstract, introduction and conclusion lack an explanation of the goal of the conducted research. It is not clear what novelty the work brings. Additionally, an abstract is not the same as a conclusion. It is enough to write in the abstract what are the main novelty and the main conclusion, and not to recount the conclusions.

I suggest that the authors do a major revision.     

References are used only in the introduction. During the article you have no referencing for exp setting, theoretical consideration, discussion of results...Use the given references or add new ones for constructive analysis.

The experimental apparatus is not listed correctly. Specify the manufacturer, the country of the manufacturer.

The figures are very bad. They are blurred and the font size is smaller than the specified text.

The main drawback of this article is the lack of a main idea why you did the research. Clearly emphasize some of your previous results that led you to this kind of research. Mention that in the introduction.

State what you got, confirmed, and what you didn't get with this kind of analysis. Elaborate on this in the discussion of the results

You should clearly state the scientific contribution of this article

Comments on the Quality of English Language

Minor editing of English language required

Author Response

Reviewer comments:

The authors described the experiments well and processed the experimental results well. In the conclusion, they summarized all the results obtained by analyzing the experimental results. However, the abstract, introduction and conclusion lack an explanation of the goal of the conducted research. It is not clear what novelty the work brings. Additionally, an abstract is not the same as a conclusion. It is enough to write in the abstract what are the main novelty and the main conclusion, and not to recount the conclusions.

Response:

Thank you for your review and valuable feedback on our paper. We will address the points you raised by revising and improving accordingly. In the revised version, We have rewritten the abstract, introducing the novelty of our study while eliminating redundant content already covered in the conclusion. Additionally, we have supplemented relevant explanations regarding the research objectives in the introduction and conclusion sections.

Comment 1:

References are used only in the introduction. During the article you have no referencing for exp setting, theoretical consideration, discussion of results...Use the given references or add new ones for constructive analysis.

Response:

Thanks for your valuable suggestions. In the revised manuscript, we have added relevant references to the theoretical analysis section.

Comment 2:

The experimental apparatus is not listed correctly. Specify the manufacturer, the country of the manufacturer.

Response:

Thanks very much for your professional suggestions. We have added relevant content to the experimental equipment and protocol section.

Comment 3:

The figures are very bad. They are blurred and the font size is smaller than the specified text.

Response:

Thank you for your valuable feedback. We have replaced unclear images in the reworked manuscript to make them more legible.

Comment 4:

The main drawback of this article is the lack of a main idea why you did the research. Clearly emphasize some of your previous results that led you to this kind of research. Mention that in the introduction.

Response:

Thank you for your valuable feedback. In our revision, We have added relevant content to the introduction section.

Comment 5:

State what you got, confirmed, and what you didn't get with this kind of analysis. Elaborate on this in the discussion of the results.

Response:

Thank you for your valuable review comments. We have added 4. Discussion to the revised manuscript and discussed the relevant content in detail.

Comment 6:

You should clearly state the scientific contribution of this article.

Response:

Thank you for your professional opinions. We have added relevant content to Section 3.3 of the revised manuscript.

Reviewer 2 Report

Comments and Suggestions for Authors

The sumbitted Manuscript entitled "Study on Characteristics of Failure and Energy Evolution of Different Moisture-Contenting Soft Rocks under Cyclic Disturbance Loading" by Xuewen Cao, Xuhui Tang, Lugen Chen, Dong Wang and Yujing Jiang presents the results of experimental investigation of the mechanical properties and energy evolution characteristics of different water-bearing soft rocks under cyclic perturbation loading. The experimental setup was composed of creeping dynamic perturbation impact loading system which was utilized to determine stress–strain curves of the experimental samples and indexes of each energy. The authors conclude, that mudstone sandstone, with a softening coefficient of 0.397, is significantly influenced by moisture content. Applied procedurÄ™ of  cyclic perturbation loading, allowed to indicate that the samples increase peak strengths to varying degrees except for the saturated rocks. Samples at 6.95% and saturation moisture states show a weakening effect, while other states exhibited a strengthening effect on the elastic modulus.

The aim and the topic of the paper is relative clear and has significant application potential. However, before the manuscript can be accepted, several improvements and updates should be applied to the paper. 

 1. Why did the Authors select the following water content in the soft rocks: 0.00%, 1.74%, 3.48%, 5.21%, 6.95% and 8.69%? Is this a result of an arbitrary drying procedure or such moisture content corresponds to some real conditions?

2. "Following the standards set by the International Society for Rock Mechanics (ISRM), standard specimens with a diameter of 50 mm and height of 100 mm were fabricated."

The Authors should provide more details concerning the sample preparation, i.e. whether the sandstone was pressurized to form the normalized specimem? 

3. "Samples, excluding those designated for drying, were immersed in distilled water. Then the samples were removed at intervals, their surface moisture was wiped off and their masses were recorded."

What intervals were applied in this procedure?

4. What do the color–shaded areas represent in Figures 3 and 5? Such information should be incorporated in the Figure caption.  

5. The values of energy density given in J/mm3 are relatively low, the Authors should consider mJ/m3 for clarity. 

6. Figure 9 – how is the damage variable D defined and what is the unit? The Authors should provide more details concerning this parameter. Furthermore, the selection of colors of data series should be updated to avoid using similar colors for different data series, e.g. blue color is used for both "dissipated energy percentage" and "elastic energy percentage". Additionally it is quite confusing if the additional blue axis corresponds to "dissipated energy percentage" or "elastic energy percentage". Left axis is labeled "Energy indicators" and it is unclear, to which data series does it match. The Authors should provide units in data series legend as the graph includes 3 vertical axes and several data series. 

In summry, the paper should be revised before it can be considered for publication. I recommend a minor revision. 

Comments on the Quality of English Language

The quality of english language is acceptable for a scientific paper.

Author Response

Comment 1:

Why did the Authors select the following water content in the soft rocks: 0.00%, 1.74%, 3.48%, 5.21%, 6.95% and 8.69%? Is this a result of an arbitrary drying procedure or such moisture content corresponds to some real conditions?

Response:

Thank you for your kind reminding. After the moisture content test experiment, the saturated moisture content of the sample was 8.69%. These moisture contents were chosen to represent 0%, 20%, 40%, 60%, 80% and 100% of the relative moisture content of the rock respectively. The moisture content is divided into smaller gradients, which correspond to the common moisture content ranges of surrounding rocks in mining environments.

Comment 2:

"Following the standards set by the International Society for Rock Mechanics (ISRM), standard specimens with a diameter of 50 mm and height of 100 mm were fabricated."

The Authors should provide more details concerning the sample preparation, i.e. whether the sandstone was pressurized to form the normalized specimem?.

Response:

Thanks very much for your professional suggestions. The preparation of standard specimens includes three processes: coring, cutting and grinding. First, use a coring machine to take out a cylinder with a diameter of 50mm from the underground surrounding rock. Control the descending speed of the drilling head to ensure that the non-verticality of the cylinder specimen is less than ±0.02mm. Cut the cylinder specimen into pieces using a rock cutting machine. The specimen that is slightly taller than the standard sample was finally polished to make both ends flat to obtain the standard specimen. During the processing, the sandstone was not additionally pressurized to form a standardized specimen. We have added relevant content in the revised manuscript.

Comment 3:

"Samples, excluding those designated for drying, were immersed in distilled water. Then the samples were removed at intervals, their surface moisture was wiped off and their masses were recorded."

What intervals were applied in this procedure?

Response:

Thank you for your kind reminding. During the experiment, we measured every 30 minutes during the first 12 hours of immersion in water, and then every 12 hours thereafter.

Comment 4:

What do the color–shaded areas represent in Figures 3 and 5? Such information should be incorporated in the Figure caption.

Response:

Thank you for your valuable feedback. The colored shaded areas in Figures 3 and 5 are intended to beautify the pictures and have no actual representative meaning. Instead, they will cause misunderstandings for readers. In the revised manuscript, we have deleted the colored shaded areas.

Comment 5:

The values of energy density given in J/mm3 are relatively low, the Authors should consider mJ/m3 for clarity.

Response:

Thank you very much for taking the time to review our manuscript. In the revised manuscript, we changed J/mm3 to mJ/m3.

Comment 6:

Figure 9 – how is the damage variable D defined and what is the unit? The Authors should provide more details concerning this parameter. Furthermore, the selection of colors of data series should be updated to avoid using similar colors for different data series, e.g. blue color is used for both "dissipated energy percentage" and "elastic energy percentage". Additionally it is quite confusing if the additional blue axis corresponds to "dissipated energy percentage" or "elastic energy percentage". Left axis is labeled "Energy indicators" and it is unclear, to which data series does it match. The Authors should provide units in data series legend as the graph includes 3 vertical axes and several data series.

Response:

Thanks very much for your professional suggestion. The definition of fatigue damage variable D usually refers to the cumulative degree of damage experienced by a material under cyclic loading. The specific calculation formula is as follows:

where:  is the initial cyclic maximum strain value,  is the instantaneous maximum strain value after n cycles, and  is the limiting strain value during the disturbance.

Since the damage variable D is calculated from the ratio of strains, it is a dimensionless quantity with no unit.

We have updated the data color in Figure 9 of the revised manuscript. The three sets of data of input energy density, elastic energy density and dissipation energy density share the left axis, are unified in black and distinguished by different symbols. The blue coordinate on the right axis represents the size of the elastic energy percentage and dissipated energy percentage data. The two share the same coordinate axis and are distinguished by different symbols. In the data legend, relevant units have been added.

Round 2

Reviewer 1 Report

Comments and Suggestions for Authors

 Accept in present form

Comments on the Quality of English Language

Minor editing of English language required